# "Dare to feel full"—A group treatment method for sustainable weight reduction in overweight and obese adults: A randomized controlled trial with 5-years follow-up

Sara Holmberg[1,2,3☯]*, Lena Lendahls[1,2☯], Kjell-Åke Alle[1,4☯]

1 Department of Research and Development, Region Kronoberg, Växjö, Sweden, 2 Faculty of Health and Life Sciences, Department of Medicine and Optometry, Linnaeus University, Växjö, Sweden, 3 Division of Occupational and Environmental Medicine, Institute of Laboratory Medicine, Lund University, Lund, Sweden, 4 Faculty of Medicine and Health Sciences, Linköping University, Linköping, Sweden

☯ These authors contributed equally to this work.
* sara.holmberg@kronoberg.se

**Data Availability Statement:** Participant consent allowed for data to be used in the specified research project. Ethical approval and data sharing regulations with protection in line with GDPR

## Abstract

### Objectives

To assess the long-term effects on weight reduction and health of a group-based behavioral weight intervention over six months focusing eating for fulfillment as compared to a control regime with brief intervention.

### Method

Overweight or obese adults (n = 176, 80% female, mean BMI 33.8 ± 4.7 kg/m$^2$, mean age 55.2 ±10.1 years) were randomized to a group treatment or control receiving a brief intervention. Ninety-three participants (53% of original sample) completed the 5-year follow-up. Anthropometrics, blood pressure and biochemical measurements, self-rated lifestyle habits, quality of life and medication were obtained at baseline, at the end of the 6-month intervention, and once a year for five years following randomization.

### Results

A per-protocol analysis, performed due to a high drop-out rate, found that weight reduction was small and similar in the two groups after five years. Reduction of waist/hip ratio, total-cholesterol and triglycerides were somewhat larger in the control group than in the treatment group. No changes regarding blood pressure, quality of life or medication use between the treatment and control groups were found.

### Conclusions

No effect on weight reduction of the group intervention was found as compared to brief intervention but both groups achieved small weight loss over time. Findings indicate that any

restricts full availability. Interested parties can obtain data for replication or for other research purposes by contacting the corresponding author Dr Sara Holmberg at sara.holmberg@kronoberg. se.

**Funding:** KÅA, LL and SH together received funding from the County Council of Kronoberg, Sweden, grant numbers 14LTK710, 15RK535, 16RK13 and the Medical Research Council of Southeast Sweden (FORSS), grant numbers FORSS-419971, FORSS-478121, FORSS-565081, FORSS-659151, FORSS-752611. The funders had no role in study design, data collection and analysis, decision to publish or preparation of the manuscript.

**Competing interests:** The authors have declared that no competing interests exist.

intervention or merely regular follow-ups might be promotive for weight maintenance in middle age.

## Introduction

Overweight and obesity are a worldwide health concern similar to an ongoing pandemic, with a considerable impact on health and quality of life. According to the World Health Organization, 59% of the adult population and one in three school-aged children are overweight or obese in Europe [1]. The Organization for Economic Co-operation and Development (OECD) calculates that the average life expectancy will decrease by three years between 2020 and 2050 due to related sequelae such as diabetes, cancer, and cardiovascular diseases [2]. The proportion with obesity has increased from 11% in 2004 to 16% in 2022 in Sweden, while the proportion with overweight and underweight has remained almost unchanged. More than half (51%) of Sweden's population between the ages of 16 and 84 reported being overweight or obese in 2022 [3].

There is scientific support for a small weight loss of 5% of the original body weight can even have beneficial effects on the health of obese people [4,5]. Maintaining a lower weight has proven more difficult as the majority begin to gain weight after 6 months [6]. Approximately 20% manage to maintain their new weight long-term [7,8]. Surgery has been the most effective treatment for severe obesity (BMI>35) [9,10], while anti-obesity medications are showing promising results [11,12] and there are several medications under development [12,13]. However, neither surgery nor medication will be suitable or affordable for all people with overweight or obesity and the side effects might hamper its use for some. Lifestyle and behavioral modifications will likely remain the cornerstones of overweight and obesity management.

Following dietary advice and engaging in physical activity correlate most strongly with weight loss [14,15]. Weight loss and weight control is a complex process that requires more than changing a single behavior and treatment needs to address not only weight loss but also overall well-being [16,17]. Person-centered care and motivational interviewing (MI) aiming to encourage people to modify habitual behavior are feasible in promoting weight loss [17–21].

Behavioral interventions targeting weight loss in individuals with overweight and obesity have demonstrated limited success in achieving desirable outcomes over both short and long-term durations. High dropout rates further exacerbate these challenges [22,23]. To overcome these limitations, it is crucial to refine existing approaches by incorporating innovative strategies that enhance participant engagement and provide tailored support. "Dare to feel full" is such a strategy with a potential to lead to a more successful outcome. The objective of this study was to investigate the long-term effects of a dietary behavioral group intervention, with an educational approach, on weight loss and health measures among overweight and obese adults.

## Methods

The study was conducted using a randomized controlled clinical intervention design with a 1:1 allocation, and the trial was registered at Clinicaltrials.gov in year 2015 number: NCT03441308. The trial was registered and approved by the Regional Ethics Review Board in Linköping, Sweden, number: 2014/231-31. All participants provided written informed consent. The implementation of the trial and the report was aimed to follow the CONSORT checklist for RCTs assessing nonpharmacological treatments [24]. The trial implementation

followed the study protocol without major changes, but the analysis was performed per-protocol instead of the intended intention-to-treat due to available data. The recruitment period started August 17, 2015 and ended September 9, 2015.

The eligibility criteria for participants included was having a body mass index (BMI) of 27–45 kg/m$^2$ based on self-reported length and weight, being 18–70 years of age, being essentially healthy (not fulfilling exclusion criteria) and having the willingness and motivation to lose weight. Exclusion criteria were severe diseases such as insulin treated diabetes mellitus, severe psychiatric disease, severe liver or kidney disease, severe heart failure, other general severe diseases, or multiple food allergies. All inclusion and exclusion criterions were based on self-report.

The study was performed in two neighboring rural counties in southern Sweden with a total population of about 450 000 inhabitants. Participants were recruited through local media and a project website. Individuals attracted to participation made a statement of interest on the project website and were contacted by telephone for screening of eligibility. The call screening was performed by an experienced district nurse.

Data was collected and the interventions were performed at three locations set up for the research purpose. All the data collection and the intervention and control regimes were performed apart from the participants' ordinary healthcare. The intervention groups were started from September 16, 2015 to November 3, 2016 and the final follow-up for data collection was in November 2021.

## Treatment intervention

The intervention under investigation denoted "Treatment" consisted of a behavioral group treatment with an educational approach over six months. Participants allocated to treatment were invited by postal mail and there were possibilities for telephone contact with the project coordinator. Ten groups of ten participants were consecutively invited to the intervention including ten sessions of two hours over six months led by the same project coordinator (an experienced district nurse). An individual session including baseline data collection was scheduled after the first group session. A short individual session for personalized counseling was offered close in time to group session five and an individual session after the final group session for collection of the 6-month follow-up data was performed. The participants were thereafter invited for a short individual session with data collection once a year after baseline for five years. The data collection was performed by the project coordinator with a few exceptions due to practical reasons.

The components of the treatment intervention "Dare to feel full" (Våga vara mätt in Swedish) had a focus on eating for satiety and normal weight. The individual target weight was discussed and determined in a dialogue at the first individual session. The group sessions consisted of talks on and discussions about food content and metabolic effects on blood sugar related to various foods, associations between food and well-being, and how to have a healthy relationship with food. Practical advice on food products, healthy and affordable food and food preparation and recipes was also provided. A common meal prepared by the project coordinator was shared at every session. No specific diet was advocated but general dietary advice on variety of food including high proportions of nutritious vegetables was recommended in accordance with general dietary recommendations. Motivational interviewing (MI) was applied as a pedagogical model [25], and all sessions included a lecture, a dialogue and group discussions. The group treatment method "Dare to feel full", in addition, used an educational plate model developed by the project coordinator that was based on the idea that each person's target weight was the basis for each person's caloric calculation, which generated the

apportioning of various foods on the individual's plate. The take-home message was to eat sufficient ordinary healthy food on a regular basis for one's target weight and when the target weight was attained to maintain eating that way for a sustainable weight.

## Control regime

Participants randomized to the control group were invited by postal mail in the same manner as the treatment group with the option of telephone contact. The participants were offered an individual counselling session, which lasted approximately one hour and was conducted by a trained nurse using MI methodology. Personalized dietary advice was provided during the session in accordance with the then existing recommendations from the Swedish Food Agency. Additionally, the control participants received the official food agency brochure [26]. The baseline data collection was performed during the counselling session. The control group participants were invited to the same nurse for a brief individual session for data collection after six months and one year from baseline and thereafter yearly, in the same manner as for the intervention group.

## Adherence

To ensure a high level of conformity and adherence from the care providers, all the intervention sessions and almost all the follow-ups in the treatment group were delivered by the same nurse. Three trained nurses provided the brief intervention and follow-up sessions in the control group. Personal continuity was sought to the greatest extent possible, and efforts were made with telephone availability and scheduling adjustments to facilitate participation and adherence to follow-up for all participants.

## Data collection procedures

The data collection procedures for this study involved seven time points, namely baseline, 6 month, 1-year, 2-year, 3-year, 4-year and 5-year follow-ups. These procedures were identical for both the intervention and control groups. Anthropometric and blood pressure measurements were taken, blood samples were collected, and a questionnaire was answered on site on each occasion. Participants were also provided with brief personalized advice and had the opportunity to ask questions.

Anthropometric measures included weight (in kg with one decimal), body mass index (BMI) (in kg per meter squared), waist and hip circumference (in centimeters) and percentage of fat mass. Weight and percentage of fat mass was measured by bioelectrical impedance analysis. Blood pressure was measured in sitting position in the right arm using a manual blood pressure cuff and recorded as systolic blood pressure (SBP) and diastolic blood pressure (DBP) in mmHg. Height (in centimeters) was only recorded at baseline. All measures were recorded on prepared papers with research person identification numbers and later transferred to an electronic database.

Blood samples collected at baseline and at all follow-ups were analyzed within 24 hours at the local hospitals' ordinary clinical laboratories (Växjö and Kalmar). The following parameters were analyzed: Hemoglobin (Hb) (g/l), Mean Corpuscular Volume (MCV) (fl), White Blood Cell Count (WBC) ($10^9$/l), Platelets (PLT) ($10^9$/l), Glycated Hemoglobin (HbA1c) (mmol/mol), Total cholesterol (TC) (mmol/l), Triglycerides (TAG) (mmol/l) and Thyroid stimulating hormone (TSH) (mU/l). All laboratory test results were assessed by one of the study's physicians. All the participants received a personal postal letter with their blood test results and explanations after each data collection session. In the case of pathological results, recommendations for action were given, and phone contact was made if medically necessary.

The baseline questionnaire included socio-economic characteristics (education and occupation), lifestyle habits, quality of life and medication. Self-rated food habits were assessed as "very good", "good", "bad" or "very bad". Other lifestyle habits assessed were tobacco use (current or previous smoking and snuffing), alcohol use and physical activity. Alcohol consumption was assessed as number of standard drinks per week and frequency of heavy drinking (four or more standard glasses for women and five for men at the same occasion). Physical activity was assessed on a five-level scale for work-related activity (sitting, standing, easier mobile work, heavier mobile work, heavy physical work) and a four-level scale for leisure time activity (sedentary, light/moderate activity, moderate regular activity and regular more intensive training). Health related quality of life was evaluated using the Gothenburg quality of life instrument including 14 items on a seven graded Likert scale [27]. Medication use was assessed as "yes" or "no" and if "yes" what kind of medication was asked for in free space. The follow-up questionnaires assessed current food habits with the same questions and an additional item on self-rated perceived change in food habits ("better", "unchanged" or "worse").

## Statistical analyses

The primary outcome was weight change (in kg) after five years follow-up. Secondary outcomes were weight change (in percent) and the proportion of participants who lost $\geq 5$ percent of body weight and weight changes within the study period. Additional secondary outcomes were percentage of fat mass, blood pressure, waist, blood lipids and quality of life. The primary analysis was designed to compare weight change after five years between the two groups. To achieve 80 percent power and an alfa level of 0.05, we calculated using Wilcoxon-Mann-Whitney test for two independent groups, a sample size based on previous studies, assuming a mean five-year weight loss of 0.5 kg in the control group and 3.0 kg in the intervention group, with a standard deviation of 5 kg. Considering a 35 percent attrition rate at five years, we enrolled 200 participants (100 per group) (G*Power 3.1). A computer-generated randomization sequence was generated using Research Randomizer (version 4.0) [28] to allocate participants into intervention or control groups. Allocation was stratified by location (county of Kronoberg and county of Kalmar) and five sets of 20 unique numbers per set were created for each location. No other restrictions were applied, and allocation was performed before the baseline and concealed in envelopes. The envelopes were revealed at the time of sending postal invitations with appointments for the first session. Neither the participants nor the intervention providers were for obvious reasons blinded to the intervention, but all the data collection and follow-ups were performed similarly for both groups over the entire five-year follow-up to minimize bias.

The results were analyzed on a per-protocol basis, according to the randomization group, due to a high drop-out rate at the five-year follow-up. The Statistical Package for the Social Sciences (IBM SPSS for Windows 2020, version 26) was used to analyze all data with a significance level set at $p < 0.05$. The normality of the data was assessed using the Shapio-Wilk test. Parametric data was presented as means ± standard deviation of the mean (SD), while non-parametric data was presented as proportions or median with interquartile range (IQR) when indicated. Descriptive statistics were used to quantify baseline characteristics. Differences between proportions were tested using the Chi-square test (with Yates' correction) and Fishers exact test, while differences between group means were tested using the non-parametric Wilcoxon and Mann-Whitney tests. P-values were subjected to Bonferroni correction to account for multiple comparisons conducted throughout the study duration.

## Results

The study recruited 245 volunteers, of which 200 participants were randomized, with 100 each in the counties of Kronoberg and Kalmar (Fig 1). The recruitment took place in 2015, and the trial was ongoing from September 2015 until November 2021. Twenty-four participants (11 in the intervention group and 13 in the control group) did not attend the first session and thus there are no baseline or outcome measures for these. Age and sex did not differ significantly between these early dropouts and the actual study group. Baseline characteristics of the 176 participants who initiated the intervention are given in Tables 1 and 2. Baseline characteristics were similar in the treatment group and control groups, except for statistically significant differences in waist circumference, self-rated eating habits, and total cholesterol. Almost two thirds of the participants used medications regularly. The most frequent medications were antihypertensives, anticoagulants and painkillers. Few participants were on antidiabetics or hypolipidemics.

Over the course of the five-year intervention, 83 participants (47%) (n = 40 in the treatment group and n = 43 in the control group) discontinued their participation for various personal reasons, with no safety-related dropouts. The non-completers were compared to the 93 participants (53%) who completed the intervention over five years (Table 3). No significant differences were found according to allocation between completers and non-completers at the different time-points except at baseline where significantly more participants in the control group did not take part (16 vs 7, p = 0.04). Non-participation at one time-point did not prevent participation at later follow-ups.

Weight changes were similar in both groups at five years (Table 4). Weight change in percent from baseline to 6 months follow-up was significantly larger in the intervention group (Median -2.11 vs -1.06, p = 0.02) but no differences were found at later follow-ups. The proportion of participants who lost ≥ 5% of body weight was also significantly higher in the intervention group at 6 months (27% vs 9%, p = 0.02) but no differences consisted at later follow-ups. Fig 2 shows the percentage of participants in the intervention group and in the control group who lost or gained weight according to 5 kg weight change categories.

Blood pressure changes did not differ significantly at any time for the treatment group compared to the control group over the study period (Table 5). Reductions of waist/hip ratio, total-cholesterol and triglycerides were larger in the control group than in the treatment group at the five-year follow-up (Table 5). TSH and HbA1c did not change between or within the two groups over the entire study period. Fat percent change according to bioimpedance measures was reduced in both groups at 6 months (n = 146, -0,69%, p = 0.024), but no differences were found between the intervention and control groups at any later time-point (Table 5).

Significant changes were found in the quality-of-life scores for the entire study group at almost all follow-ups as compared to baseline but no difference between the treatment group and the control group were seen at any time-point (Table 6). No significant changes in medication were found from baseline to five-year follow-up in any of the medication groups investigated (hypertension, diabetes, asthma, pain, heart failure and metabolism), and no differences between the two groups were revealed. No adverse effects related to the intervention or control regime occurred, and no dropouts reported adverse effect as the reason for discontinuing participation.

## Discussion

The present randomized trial found no effect on weight reduction of the treatment intervention, a behavioral education method over six months, when compared to a brief intervention among overweight or obese adults. However, both groups had a small and similar average

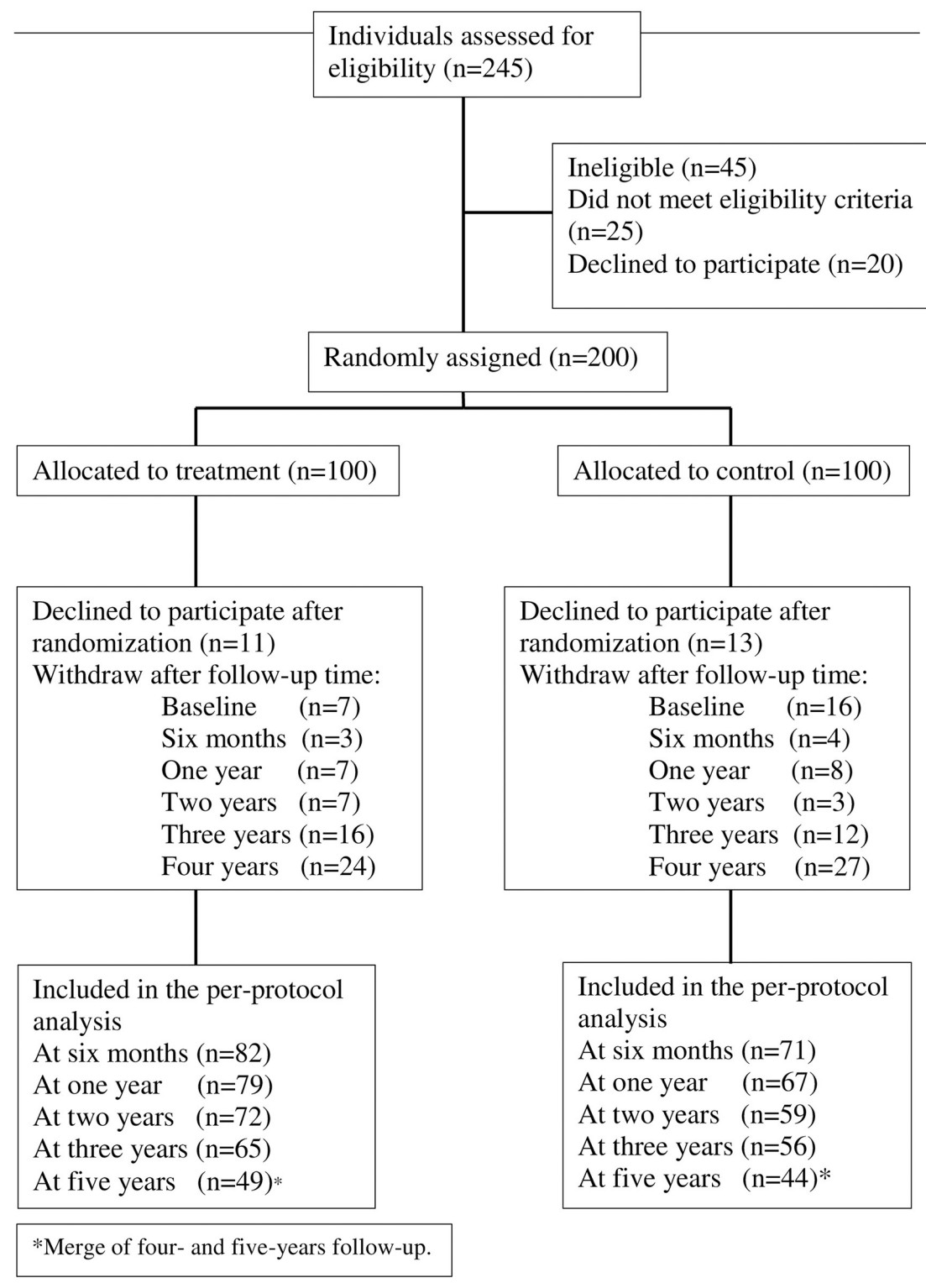

**Fig 1. Flow diagram of the study population.**

**Table 1. Baseline characteristics of all patients (n = 176).**

| | All (n = 176) | Treatment (n = 89) | Control (n = 87) |
|---|---|---|---|
| Male/Female (n/n) | 36/140 | 14/75 | 22/65 |
| | Median (IQR) | Median (IQR) | Median (IQR) |
| Age (year) | 56.5 (12.3) | 55.0 (13.0) | 57.0 (12.5) |
| Weight (kg) | 93.6 (20.7) | 95.3 (20.1) | 92.4 (20.5) |
| Height (cm) | 168.0 (11.0) | 167.0 (9.0) | 168.0 (13.0) |
| Body mass index (kg/m$^2$) | 32.8 (5.2) | 33.2 (4.5) | 32.4 (5.7) |
| Fat Mass (%) | 43.7 (7.4) | 44.5 (7.6) | 43.3 (7.4) |
| Hip (cm) | 118.0 (11.0) | 118.0 (9.0) | 117.5 (12.5) |
| Waist (cm)* | 104.0 (13.0) | 101.0 (12.0) | 107.0 (11.5) |
| SBP[#] (mmHg) | 135.0 (10.5) | 130.0 (10.0) | 135.0 (11.0) |
| DBP[$] (mmHg) | 80.0 (5.0) | 85.0 (10.0) | 80.0 (10.0) |
| Laboratory data[b] | Median (IQR); n | Median (IQR); n | Median (IQR); n |
| Hb (g/l) | 139.0 (13.0); 157 | 138.0 (12.0); 85 | 140.5 (14.3); 72 |
| MCV (fl) | 90.0 (6.0); 155 | 90 (4.3); 84 | 90 (6.0); 71 |
| WBC (10$^9$/l) | 7.1 (2.4); 155 | 7.3 (2.3); 84 | 7 (2.6); 71 |
| PLT (10$^9$/l) | 263.0 (87.5); 155 | 257 (93.8); 84 | 264 (66.0); 71 |
| HbA1c (mmol/mol) | 37.0 (5.0); 158 | 37.0 (5.0); 87 | 37.0 (4.5); 71 |
| TC (mmol/l)* | 5.6 (1.7); 159 | 5.2 (1.5); 87 | 6.0 (1.7); 72 |
| TAG (mmol/l) | 1.9 (1.3); 156 | 1.9 (1.2); 85 | 1.9 (1.3); 71 |
| TSH (mU/l) | 1.7 (1.4); 159 | 1.8 (1.4); 87 | 1.6 (1.3); 72 |

*Statistically significant difference at baseline between treatment and control (Mann-Whitney).

[#]Systolic blood pressure.

[$]Diastolic blood pressure.

[b]Hemoglobin (Hb) (g/l), Mean Corpuscular Volume (MCV) (fl), White Blood Cell Count (WBC) (10$^9$/l), Platelets (PLT) (10$^9$/l), Glycated Hemoglobin (HbA1c) (mmol/mol), Total cholesterol (TC) (mmol/l), Triglycerides (TAG) (mmol/l) and Thyroid stimulating hormone (TSH) (mU/l).

weight loss over time. These findings indicate that none of the interventions had a clinically meaningful impact on weight loss or metabolic health but possibly on weight maintenance at group level.

The initial weight loss observed in both groups can partly be attributed to the components of the placebo effect provided during the six-month intervention, as reported in previous studies [29,30]. However, this initial effect did not translate into sustained long-term weight loss, as there was a pattern of weight regain following the initial loss which occurred within the first six months. This pattern of weight loss and regain is consistent with evidence from several other randomized trials which have also demonstrated a high rate of weight regain beyond the one year [7,31–33]. Previous research has indicated that group attendance significantly predicts weight loss success in weight control programs for adolescents [34], but this was not observed in the present study with adults. This discrepancy may be attributed to differences in the study population, or the specific interventions employed. The lack of clinically relevant improvements observed in this study was also evident for the measured metabolic risk factors, including waist circumference, blood pressure, HbA1c and lipid levels. It is worth mentioning that even modest weight loss of as little as 5% of the original body weight can have beneficial health effects [4,5]. However, maintaining the reduced weight has proven to be challenging. This study might support that regular follow-up with a person-centered approach and continuity of care may promote weight maintenance although there was significant variation within

**Table 2.  Baseline characteristics of all patients, demographic data (n = 176).**

| | | All | Treatment | Control |
|---|---|---|---|---|
| **Employment** | | n (%) | n (%) | n (%) |
| | Full-time | 84 (47.7) | 44 (49.4) | 40 (46) |
| | Part-time | 41 (23.3) | 18 (20.2) | 23 (26.4) |
| | Student | 3 (1.7) | 2 (2.2) | 1 (1.1) |
| | Sick leave >3 months | 5 (2.8) | 5 (5.6) | 0 (0) |
| | Parental leave | 0 (0) | 0 (0) | 0 (0) |
| | Pensioner | 36 (20.5) | 16 (18.0) | 20 (23) |
| | Other | 7 (4.0) | 4 (4.5) | 3 (3.4) |
| **Profession** | | | | |
| | Care/School | 53 (36.1) | 22 (29.3) | 31 (43.1) |
| | Industry/Agriculture | 16 (10.9) | 11 (14.7) | 5 (6.9) |
| | Office/Trade | 63 (42.9) | 34 (45.3) | 29 (40.3) |
| | Other | 15 (10.2) | 8 (10.7) | 7 (9.7) |
| **Education** | | | | |
| | Compulsory school | 21 (12.1) | 12 (13.6) | 9 (10.5) |
| | Upper secondary school | 71 (40.8) | 38 (43.2) | 33 (38.4) |
| | College | 63 (36.2) | 34 (38.6) | 29 (33.7) |
| | Other | 19 (10.9) | 4 (4.5) | 15 (17.4) |
| **Medication use** | | | | |
| | No | 38 (22.1) | 22 (24.7) | 16 (19.3) |
| | Yes, regularly | 110 (64.0) | 55 (61.8) | 55 (66.3) |
| | Yes, if necessary | 24 (14.0) | 12 (13.5) | 12 (14.5) |
| **Eating habits*** | | | | |
| | Very good | 1 (0.6) | 0 (0) | 1 (1.2) |
| | Good | 102 (60.4) | 40 (48.2) | 62 (72.1) |
| | Bad | 55 (32.5) | 33 (39.8) | 22 (25.6) |
| | Very bad | 11 (6.5) | 10 (12) | 1 (1.2) |
| **Tobacco habits** | | | | |
| | Never smoked or snuffed | 88 (50.0) | 50 (56.2) | 38 (43.7) |
| | Stopped smoking or snuff | 62 (35.2) | 28 (31.5) | 34 (39.1) |
| | Yes, but not daily | 4 (2.3) | 1 (1.1) | 3 (3.4) |
| | Smokes daily | 7 (4.0) | 4 (4.5) | 3 (3.4) |
| | Snuff daily | 10 (5.7) | 4 (4.5) | 6 (6.9) |
| | Other | 5 (2.8) | 2 (2.2) | 3 (3.4) |
| **Alcohol habits** | | | | |
| | Sober | 21 (11.9) | 12 (13.5) | 9 (10.3) |
| | <1 standard glass/week | 63 (35.8) | 32 (36.0) | 31 (35.6) |
| | 1–4 standard glass/week | 61 (34.7) | 31 (34.8) | 30 (34.5) |
| | 5–9 standard glass/week | 23 (13.1) | 10 (11.2) | 13 (14.9) |
| | 10–14 standard glass/week | 7 (4.0) | 3 (3.4) | 4 (4.6) |
| | >15 standard glass/week | 1 (0.6) | 1 (1.1) | 0 (0) |
| **Physical activity work** | | | | |
| | Sedentary | 62 (36.5) | 31 (35.2) | 31 (37.8) |
| | Standing | 16 (9.4) | 11 (12.5) | 5 (6.1) |
| | More easily mobile | 70 (41.2) | 35 (39.8) | 35 (42.7) |
| | Heavier mobile | 20 (11.8) | 9 (10.2) | 11 (13.4) |
| | Heavy | 2 (1.2) | 2 (2.3) | 0 (0) |

(*Continued*)

**Table 2.** (Continued)

|  |  | **All** | **Treatment** | **Control** |
|---|---|---|---|---|
| **Physical activity leisure time** |  |  |  |  |
|  | Sitting | 24 (13.8) | 11 (12.5) | 13 (15.1) |
|  | Light/moderate exercise | 97 (55.7) | 50 (56.8) | 47 (54.7) |
|  | Moderate/regular exercise | 44 (25.3) | 25 (28.4) | 19 (22.1) |
|  | Intense regular exercise | 9 (5.2) | 2 (2.3) | 7 (8.1) |

*Statistically significant difference at baseline between treatment and control (Mann-Whitney).

the population. Continuity of care has been found to improve outcomes in other chronic diseases [35,36]. The impact of continuity of care merits further investigation with regard to obesity as a chronic medical entity seriously impacting individual health.

Only 14 individuals of the initial cohort of 176 participants achieved a clinically significant weight loss ($\geq$ 5%) irrespective of their group assignment. This finding suggests suboptimal adherence to the prescribed interventions, consistent with the majority of studies on diet and weight reduction [37]. However, considering the continuous weight gain observed in the Swedish population over the last decades [38], our study implies that both treatment options

**Table 3.** Baseline characteristics of completers and non-completers.

|  | **Completers (n = 93)** | **Non-completers (n = 83)** |
|---|---|---|
| **Treatment / Control (n (%))** | 49 (52.7) / 44 (47.3) | 40 (48.2) /43 (51.8) |
| **Male / Female (n (%))** | 19 (20.4) / 74 (79.6) | 17 (20.5) / 66 (79.5) |
| **Variables*** | **Median (IQR); (n)** | **Median (IQR); (n)** |
| **Age (years)** | 58.0 (13.0); (93) | 55.0 (12.0); (83) |
| **Weight (kg)** | 92.0 (17.4); (93) | 97.1 (20.8); (83) |
| **Height (cm)** | 167.0 (11.0); (93) | 168.0 (11.0); (83) |
| **BMI (kg/m$^2$)** | 32.7 (5.1); (93) | 32.8 (5.9); (83) |
| **Waist (cm)** | 103.0 (13.5); (93) | 106.0 (11.8); (83) |
| **Hip (cm)** | 117.5 (10.0); 93 | 119.0 (11.8); (83) |
| **Fat Mass (%)** | 43.2 (7.3); (93) | 44.5 (6.2); (83) |
| **SBP (mmHg)** | 135.0 (15.0); (93) | 135.0 (11.0); (83) |
| **DBP (mmHg)** | 80.0 (5.0); (93) | 84.0 (10.0); (83) |
| **Hb (g/l)** | 139.0 (14.0); (91) | 139.0 (14.0); (65) |
| **MCV (fl)#** | 90.0 (5.0); (89) | 92.0 (5.0); (65) |
| **WBC (10$^9$/l)#** | 6.9 (2.0); (89) | 7.5 (2.7); (65) |
| **PLT (10$^9$/l)** | 255.0 (71.0); (89) | 273.0 (93.0); (65) |
| **HbA1c (mmol/mol)** | 37.0 (5.5); (91) | 37.0 (4.8); (66) |
| **TC (mmol/l)** | 5.6 (1.6); (92) | 5.7 (1.6); (66) |
| **TAG (mmol/l)** | 1.8 (1.2); (90) | 1.9 (1.3); (65) |
| **TSH (mU/l)** | 1.8 (1.3); (92) | 1.6 (1.5); (66) |

*Body mass index (BMI) (kg/m$^2$), Systolic blood pressure (SBP), Diastolic blood pressure (DBP), Hemoglobin (Hb) (g/l), Mean Corpuscular Volume (MCV) (fl), White Blood Cell Count (WBC) (10$^9$/l), Platelets (PLT) (10$^9$/l), Glycated Hemoglobin (HbA1c) (mmol/mol), Total cholesterol (TC) (mmol/l), Triglycerides (TAG) (mmol/l) and Thyroid stimulating hormone (TSH) (mU/l).
#Statistically significant difference at baseline between completers and non-completers (Mann-Whitney).

**Table 4. Median changes in primary and secondary weight-related outcomes (n = 176).**

| | Treatment | Control | p* |
|---|---|---|---|
| **Weight change in kg over follow-up** | **Median (IQR); n** | **Median (IQR); n** | |
| **Change at six months** | -1.95 (4.95); 82 | -1.00 (3.30); 66 | 0.02 |
| **Change at one year** | -1.60 (5.30); 77 | -1.00 (4.60); 65 | 1.00 |
| **Change at two years** | -0.95 (7.40); 70 | -1.00 (7.23); 54 | 1.00 |
| **Change at five years#** | -0.50 (6.50); 49 | -0.65 (9.00); 44 | 1.00 |
| **Weight change in %** | **Median (IQR); n** | **Median (IQR); n** | |
| **At six months** | -2.11 (5.05); 82 | -1.06 (3.48); 66 | 0.02 |
| **At one year** | -1.60 (5.31); 77 | -1.21 (4.54); 65 | 1.00 |
| **At two years** | -0.97 (7.60); 70 | -1.29 (7.09); 54 | 1.00 |
| **At five years#** | -0.56 (7.48); 49 | -0.74 (10.15); 44 | 1.00 |
| **Reduction of > 5% of body weight** | **Proportion; %** | **Proportion; %** | |
| **At six months** | 22/82; (27) | 6/66; (9) | 0.03 |
| **At one year** | 17/77; (22) | 12/65; (19) | 1.00 |
| **At two years** | 15/70; (21) | 15/54; (28) | 1.00 |
| **At five years#** | 13/49; (27) | 12/44; (30) | 1.00 |

*Mann-Whitney test for comparison between groups and Fisher's Exact test for comparison between proportions. Bonferroni adjusted for multiple comparisons.

#Merge of four- and five-years follow-up.

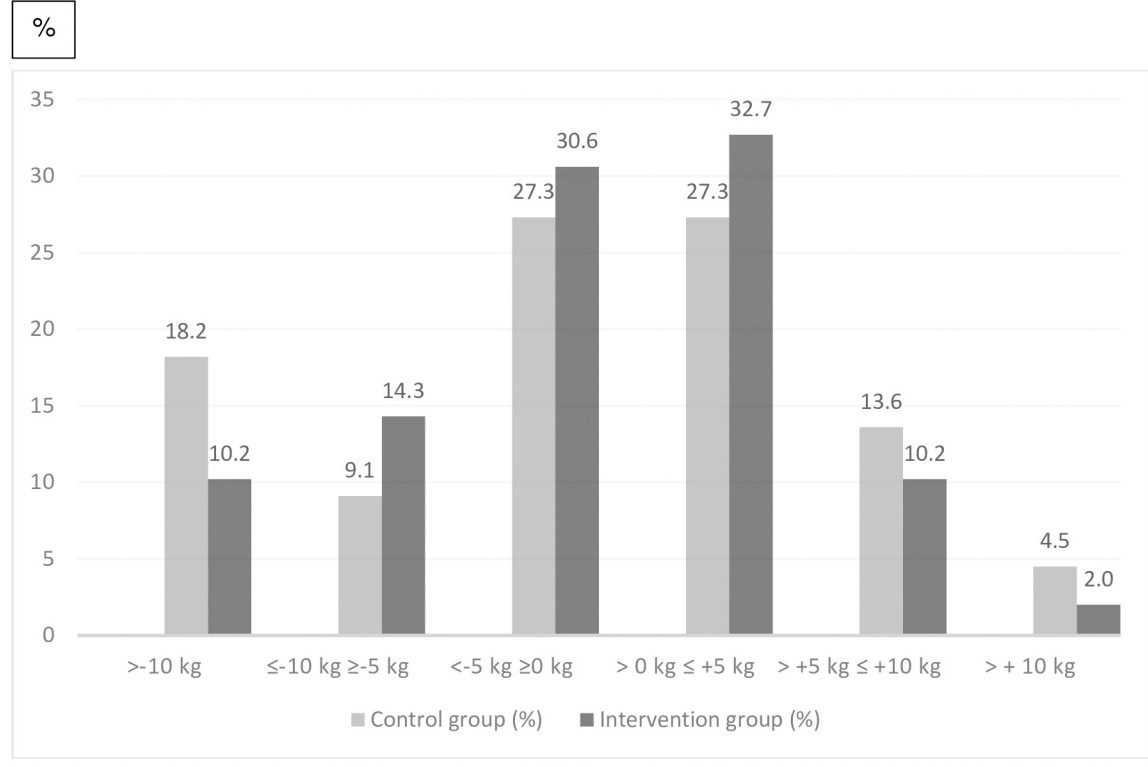

**Fig 2. Percentages of participants in the study (control group and intervention group) who met various weight loss categories at five-year follow-up.**

**Table 5. Mean change the secondary outcomes; blood pressure, waist, lipids and fat percentage between six months to five years as compared to baseline.**

| Variables* | | Six months | | | One year | | | Two years | | | Five years§ | | |
|---|---|---|---|---|---|---|---|---|---|---|---|---|---|
| | | Mean (SD) | n | p# | Mean (SD) | N | p# | Mean (SD) | n | p# | Mean (SD) | N | p# |
| **SBP (mm Hg)** | | | | | | | | | | | | | |
| | **All** | -3.51 (12.11) | 148 | | -3.01 (13.90) | 142 | | -2.64 (13.7) | 125 | | -2.10 (15.09) | 93 | |
| | **Treatment** | -4.92 (11.00) | 82 | 0.40 | -2.78 (13.07) | 77 | 1.00 | -2.89 (12.17) | 71 | 1.00 | -4.70 (14.25) | 49 | 0.28 |
| | **Control** | -1.77 (13.23) | 66 | | -3.29 (14.94) | 65 | | -2.32 (15.74) | 54 | | 0.80 (15.62) | 44 | |
| **DBP (mm Hg)** | | | | | | | | | | | | | |
| | **All** | -2.10 (6.72) | 148 | | -2.18 (7.70) | 142 | | -2.47 (7.76) | 125 | | -1.22 (9.63) | 93 | |
| | **Treatment** | -3.11 (5.50) | 82 | 0.21 | -2.48 (6.87) | 77 | 0.92 | -2.39 (6.9) | 71 | 1.00 | -3.06 (9.36) | 49 | 0.12 |
| | **Control** | -0.85 (7.84) | 66 | | -1.82 (8.61) | 65 | | -2.57 (8.84) | 54 | | 0.84 (9.61) | 44 | |
| **Waist (cm)** | | | | | | | | | | | | | |
| | **All** | -2.39 (3.75) | 148 | | -2.28 (4.74) | 142 | | -1.17 (5.72) | 125 | | -0.33 (9.16) | 93 | |
| | **Treatment** | -2.34 (3.74) | 82 | 1.00 | -1.99 (4.15) | 77 | 1.00 | 0.035 (5.06) | 71 | 0.04 | 1.72 (6.87) | 49 | 0.08 |
| | **Control** | -2.46 (3.79) | 66 | | -2.61 (5.37) | 65 | | -2.76 (6.16) | 54 | | -2.62 (8.91) | 44 | |
| **Waist/Hip ratio** | | | | | | | | | | | | | |
| | **All** | -0.005 (0.03) | 148 | | -0.006 (0.03) | 142 | | 0.003 (0.04) | 125 | | 0.007 (0.05) | 92 | |
| | **Treatment** | -0.002 (0.03) | 82 | 1.00 | -0.004 (0.02) | 77 | 1.00 | 0.010 (0.03) | 71 | 0.02 | 0.019 (0.04) | 49 | 0.01 |
| | **Control** | -0.008 (0.03) | 66 | | -0.009 (0.03) | 65 | | -0.005 (0.04) | 54 | | -0.008 (0.05) | 43 | |
| **TC (mmol/l)** | | | | | | | | | | | | | |
| | **All** | -0.04 (0.76) | 145 | | -0.02 (0.84) | 139 | | -0.03 (0.92) | 120 | | -0.19 (1.19) | 89 | |
| | **Treatment** | 0.03 (0.77) | 80 | 1.00 | 0.19 (0.78) | 75 | 0.01 | 0.21 (0.67) | 68 | 0.01 | 0.10 (1.07) | 48 | 0.01 |
| | **Control** | -0.13 (0.75) | 65 | | -0.25 (0.85) | 64 | | -0.36 (1.09) | 52 | | -0.53 (1.25) | 41 | |
| **TAG (mmol/l)** | | | | | | | | | | | | | |
| | **All** | -0.15 (0.92) | 143 | | -0.10 (1.09) | 135 | | -0.15 (0.90) | 118 | | -0.04 (1.11) | 88 | |
| | **Treatment** | 0.01 (0.74) | 79 | 0.11 | -0.01 (1.18) | 72 | 1.00 | 0.02 (0.73) | 67 | 0.43 | 0.28 (0.93) | 47 | 0.01 |
| | **Control** | -0.34 (1.06) | 64 | | -0.21 (1.05) | 63 | | -0.36 (1.05) | 51 | | -0.40 (1.20) | 41 | |
| **Fat % change** | | | | | | | | | | | | | |
| | **All** | -0.69 (2.31) | 146 | 0.02 | -0.41 (2.86) | 140 | 0.52 | -0.23 (2.94) | 120 | 1.00 | -0.38 (3.22) | 85 | 1.00 |
| | **Treatment** | -0.90 (2.04) | 80 | 0.80 | 0.33 (2.38) | 75 | 1.00 | 0.13 (2.38) | 66 | 0.24 | -0.04 (3.12) | 46 | 1.00 |
| | **Control** | -0.44 (2.60) | 66 | | -0.51 (3.34) | 65 | | -0.68 (3.49) | 54 | | -0.79 (3.33) | 39 | |

*Systolic blood pressure (SBP) (mmHg), Diastolic blood pressure (DBP) (mmHg), Total cholesterol (TC) (mmol/l), Triglycerides (TAG) (mmol/l).

§Merge of four and five-yeas follow-up.

#Bonferroni adjusted for multiple comparisons.

might have some effect in counteracting further weight gain. However, the lack of a population-based control group free of any intervention, not even taking part in data collection, hinders quantification of this potential effect. Furthermore, it is essential to consider that the final two years of the study coincided with the Covid-19 pandemic, which likely impacted

**Table 6. Change in Quality of life (QoL) from six month to five years follow-up compared to baseline.**

| | | Six months | | | One year | | | Two years | | | Five years* | | |
|---|---|---|---|---|---|---|---|---|---|---|---|---|---|
| | | Mean (SD) | N | p# | Mean (SD) | n | p# | Mean (SD) | n | p# | Mean (SD) | n | p# |
| **QoL score change** | All | 4.2 (8.2) | 107 | 0.01 | 2.0 (7.8) | 111 | 0.02 | 1.9 (8.8) | 99 | 0.07 | 4.2 (9.8) | 76 | 0.01 |
| | Treatment | 5.1 (8.7) | 61 | 0.60 | 2.3 (7.6) | 63 | 1.00 | 0.9 (9.1) | 61 | 1.00 | 5.0 (9.8) | 43 | 1.00 |
| | Control | 3.0 (7.4) | 46 | | 1.6 (7.8) | 48 | | 3.4 (8.2) | 38 | | 3.1 (9.8) | 33 | |

*Merge of four and five-year follow up.

#Bonferroni adjusted for multiple comparisons.

compliance and participation rates. Data collection and follow-ups were paused for one year, and the pandemic may have hindered participants' ability to fully engage with the interventions, potentially influencing the observed outcomes.

While surgical interventions have proven effective treatment for severe obesity [39], they are not suitable for all individuals. Anti-obesity medication has recently shown [13] promising outcomes with respect to weight reduction, and ongoing research is focused on developing new drugs. However, investigations into the consequences of discontinuing once-weekly subcutaneous semaglutide treatment and implementing lifestyle interventions have revealed that, one year later, participants tend to regain approximately two-thirds of their initial weight loss. Additionally, their cardiometabolic variables exhibited similar trends, as ascertained by Wilding et al. 2022 [40]. Neither surgery nor medication will be appropriate and affordable methods for all those with overweight or obesity. It is therefore likely that lifestyle and behavioral modifications will be crucial in secondary prevention and management of overweight and obesity also in the future. However, the probability of reaching normal weight or maintaining substantial weight loss among adults with overweight or obesity was found to be very low when patients receiving bariatric surgery were excluded [41,42]. The methods available to enhance behavioral and lifestyle changes evidently needs innovation to improve in both clinical and public health contexts.

## Strengths and limitations

The strengths of the current study include its randomized controlled design, relatively large sample size, and thorough and repeated assessment of weight and anthropometrics over a five-year period. The use of a single specially educated nurse for the treatment group and three educated nurses for the control group provided continuity and consistency. However, outcomes in both groups may have been influenced not only by the interventions themselves but also by the four different nurses delivering the interventions. The high attrition rate introduces potential for bias and may impact the generalizability of the findings. Additionally, the use of per-protocol analysis instead of intention-to-treat analysis may have slightly overestimated adherence to the interventions, as individuals who withdrew from the study were not accounted for. Moreover, the implementation of imputation as a statistical method to estimate missing data was contemplated; however, due to a substantial number of missing variables and repeated time-points, this approach may introduce uncertainty, bias, complexity and ignorable missingness, rendering it unsuitable for our data structure [43]. The p-value threshold of 0.05 has limitations when multiple comparisons or repeated measurements are performed, leading to an increased risk of false positives. We used Bonferroni corrected p-values and non-parametric statistics which can help mitigate these issues associated with multiple comparisons.

## Conclusion

In conclusion, the present trial found that the behavioral education method "Dare to feel full" was no better than a brief intervention regarding effect on weight loss or several cardiovascular disease risk factors. The high attrition rate and observed weight regain in the study suggest the need for alternative strategies. Further research is warranted to identify and develop effective and safe methods for sustaining weight loss among obese individuals and ultimately prevent obesity on a societal level.

## Supporting information

**S1 Checklist. CONSORT 2010 checklist of information to include when reporting a randomised trial\*.**
(DOC)

**S1 File.**
(DOCX)

**S2 File.**
(DOCX)

## Acknowledgments

The authors express their gratitude to all participants in the study and to participating staff. A special thanks to Mari Bergenholtz, registered district nurse, for project coordination and initiating the study idea.

## Author Contributions

**Conceptualization:** Sara Holmberg, Lena Lendahls, Kjell-Åke Alle.

**Data curation:** Sara Holmberg, Lena Lendahls, Kjell-Åke Alle.

**Formal analysis:** Sara Holmberg, Lena Lendahls, Kjell-Åke Alle.

**Funding acquisition:** Sara Holmberg, Lena Lendahls, Kjell-Åke Alle.

**Investigation:** Sara Holmberg, Lena Lendahls, Kjell-Åke Alle.

**Methodology:** Sara Holmberg, Lena Lendahls, Kjell-Åke Alle.

**Project administration:** Sara Holmberg, Lena Lendahls, Kjell-Åke Alle.

**Resources:** Sara Holmberg, Lena Lendahls, Kjell-Åke Alle.

**Software:** Sara Holmberg, Lena Lendahls, Kjell-Åke Alle.

**Supervision:** Sara Holmberg, Lena Lendahls, Kjell-Åke Alle.

**Validation:** Sara Holmberg, Lena Lendahls, Kjell-Åke Alle.

**Visualization:** Sara Holmberg, Lena Lendahls, Kjell-Åke Alle.

**Writing – original draft:** Sara Holmberg, Lena Lendahls, Kjell-Åke Alle.

**Writing – review & editing:** Sara Holmberg, Lena Lendahls, Kjell-Åke Alle.

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
