## [Decision Letter · Decision Letter 0]

24 Jan 2024

PONE-D-23-38830“Dare to feel full” - A group treatment method for sustainable weight reduction in overweight and obese adults: A randomized controlled trial with 5-years follow-upPLOS ONE

Dear Dr. Holmberg,

Thank you for submitting your manuscript to PLOS ONE. After careful consideration, we feel that it has merit but does not fully meet PLOS ONE’s publication criteria as it currently stands. Therefore, we invite you to submit a revised version of the manuscript that addresses the points raised during the review process.

We look forward to receiving your revised manuscript.

Kind regards,

Aleksandra Klisic

Academic Editor

PLOS ONE

Journal Requirements:

Reviewers' comments:

Reviewer's Responses to Questions

**Comments to the Author**

1. Is the manuscript technically sound, and do the data support the conclusions?

Reviewer #1: Partly

Reviewer #2: Yes

Reviewer #3: Yes

2. Has the statistical analysis been performed appropriately and rigorously? 

Reviewer #1: No

Reviewer #2: Yes

Reviewer #3: Yes

3. Have the authors made all data underlying the findings in their manuscript fully available?

Reviewer #1: No

Reviewer #2: Yes

Reviewer #3: Yes

4. Is the manuscript presented in an intelligible fashion and written in standard English?

Reviewer #1: Yes

Reviewer #2: Yes

Reviewer #3: Yes

5. Review Comments to the Author

Reviewer #1: 1. Clarify what test was used for sample size calculation. Better use a method for repeated measures.

2. The statistical analysis is not acceptable. For a longitudinal study with repeated measures, better consider longitudinal models such as mixed model and GLIMMIX models.

3. Specify primary endpoint (e.g. >5% reduction) and secondary endpoint. P value can be adjusted for secondary endpoint so power will not be reduced.

Reviewer #2: The study titled "“Dare to feel full” - A group treatment method for sustainable weight reduction in overweight and obese adults: A randomized controlled trial with 5-years follow-up" is well written. I have some comments for improvement:

1- Define abbreviations used in tables/figures in the figure caption.

2- Mention the strengths and limitations of your study in a separate paragraph before the conclusions paragraph.

3- Add the clinical relevance of your study.

4- Add a new heading for conclusions paragraph.

Reviewer #3: Main remarks:

• Authors should specify the randomization process.

• How is the eligibility criterion “being essentially healthy” is defined?

• Some tables (e.g. table 4, tables 7-8) should be moved to the supplementary appendix.

• Could you provide in detail the classes of drugs taken “regularly” by 64% of enrolled subjects, according to data provided in table 2 (antihypertensives, hypolipidemic, antidiabetics?)?

• Despite the absence of an effect on HbA1c levels after 5 years of follow-up in both treatment arms, it would be interesting to know how many participants developed type 2 diabetes during study course.

6. PLOS authors have the option to publish the peer review history of their article (what does this mean?). If published, this will include your full peer review and any attached files.

Reviewer #1: No

Reviewer #2: No

Reviewer #3: No

---

## [Author Response · Author response to Decision Letter 0]

1 Mar 2024

Response to reviewers

We thank the reviewers for reading critically and thereby contributing to improvement of our manuscript. Please find our response point-by-point below.

Reviewer #1 

1. Clarify what test was used for sample size calculation. Better use a method for repeated measures.

 Answer:We have added the test name in the method section.

2. The statistical analysis is not acceptable. For a longitudinal study with repeated measures, better consider longitudinal models such as mixed model and GLIMMIX models. 

Answer:We consider our statistical analysis the most appropriate given the dataset, the size of the data material and the drop-out rate. The proposed methods would render estimated data results and we argue that such method would not alter the main results and it would not give more accuracy to the analysis.

3. Specify primary endpoint (e.g. >5% reduction) and secondary endpoint. P value can be adjusted for secondary endpoint so power will not be reduced. 

Answer: Primary outcome is already specified under statistical analysis in the method section. We have added all the additional secondary outcomes in the same place in the text.

All p-values were repeated analysis were performed have been adjusted with Bonferroni correction. 

Reviewer #2 

The study is well written 

1. Define abbreviations used in tables/figures in the figure caption. 

Answer: We have excluded abbreviations from table 1. In table 3, 4 and 7 we have added definition of abbreviations in footnotes. 

2. Mention the strengths and limitations of your study in a separate paragraph before the conclusions paragraph.

 Answer: We have moved the paragraph about strengths and limitations of our study to just before the conclusion. We added a sub-heading “Strengths and limitations”.

3. Add the clinical relevance of your study.

 Answer: The clinical relevance of our findings is already presented in the last sentence of the first paragraph of the discussion and in the last five rows in the second paragraph of the discussion. We do not consider it relevant to elaborate further on this issue given the aim of the study.

4. Add a new heading for conclusions paragraph. 

Answer: We have added the suggested sub-heading.

Reviewer #3 

Specify the randomization process 

Answer: We have clarified the writing on the randomization in the method section.

How is the eligibility criterion “being essentially healthy” is defined? 

Answer: The definition was “not fulfilling exclusion criteria” and this has been added in a parenthesis.

Some tables (e.g. table 4, tables 7-8) should be moved to the supplementary appendix 

Answer: We disagree with the proposal. We prefer to keep all the tables in the main manuscript. Regarding table 4 we consider these results important for the evaluation of the study due to the high drop-out rate. Table 7 and 8 reports some of the secondary outcomes and we consider the secondary outcomes equally relevant.

Could you provide in detail the classes of drugs taken “regularly” by 64% of enrolled subjects, according to data provided in table 2 (antihypertensives, hypolipidemic, antidiabetics?)?

 Answer: We have provided brief information on type of medication in the first paragraph in the result section.

Despite the absence of an effect on HbA1c levels after 5 years of follow-up in both treatment arms, it would be interesting to know how many participants developed type 2 diabetes during study course. 

Answer: We do not have access to complete diagnostic data or diagnoses within ordinary health care. Although of clinical interest data to answer this question was not part of our data collection.

---

## [Decision Letter · Decision Letter 1]

12 Mar 2024

PONE-D-23-38830R1“Dare to feel full” - A group treatment method for sustainable weight reduction in overweight and obese adults: A randomized controlled trial with 5-years follow-upPLOS ONE

Dear Dr. Holmberg,

Thank you for submitting your manuscript to PLOS ONE. After careful consideration, we feel that it has merit but does not fully meet PLOS ONE’s publication criteria as it currently stands. Therefore, we invite you to submit a revised version of the manuscript that addresses the points raised during the review process.

We look forward to receiving your revised manuscript.

Kind regards,

Aleksandra Klisic

Academic Editor

PLOS ONE

Journal Requirements:

Reviewers' comments:

Reviewer's Responses to Questions

**Comments to the Author**

1. If the authors have adequately addressed your comments raised in a previous round of review and you feel that this manuscript is now acceptable for publication, you may indicate that here to bypass the “Comments to the Author” section, enter your conflict of interest statement in the “Confidential to Editor” section, and submit your "Accept" recommendation.

Reviewer #1: (No Response)

Reviewer #2: All comments have been addressed

Reviewer #3: (No Response)

2. Is the manuscript technically sound, and do the data support the conclusions?

Reviewer #1: (No Response)

Reviewer #2: (No Response)

Reviewer #3: (No Response)

3. Has the statistical analysis been performed appropriately and rigorously? 

Reviewer #1: (No Response)

Reviewer #2: (No Response)

Reviewer #3: (No Response)

4. Have the authors made all data underlying the findings in their manuscript fully available?

Reviewer #1: (No Response)

Reviewer #2: (No Response)

Reviewer #3: (No Response)

5. Is the manuscript presented in an intelligible fashion and written in standard English?

Reviewer #1: (No Response)

Reviewer #2: (No Response)

Reviewer #3: (No Response)

6. Review Comments to the Author

Reviewer #1: Tables 1-3 can be combined. If a nonparametric method is used, better report median (IQR) instead of mean (SD).

Table 4: this table can be omitted or moved to supplemental. If a nonparametric method is used, better report median (IQR) instead of mean (SD). Add percentages to treatment and sex. MCV superscript can be removed. Add “Abbreviations” to the footnote.

Table 5: Why sample sizes are different for weight change, % change and reduction >=5% in control? E.g. 6m, 2 yr..

Table 5: my calculation for the p value of reduction >=2% is 0.01. Are other pvalues accurate?

Weight % change and >=5% are secondary endpoint and p values may need to be adjusted. They can be presented in the table with other secondary endpoints.

Tables 6 and 7 can be combined.

Figure 2. bars are overlapping between two groups.

Reviewer #2: (No Response)

Reviewer #3: I would like to thank the authors for adequately responding to all comments and performing appropriate amendments in the manuscript.

7. PLOS authors have the option to publish the peer review history of their article (what does this mean?). If published, this will include your full peer review and any attached files.

Reviewer #1: No

Reviewer #2: No

Reviewer #3: No

---

## [Author Response · Author response to Decision Letter 1]

8 Apr 2024

We are thankful to all the reviewers for valuable comments on our manuscript and we have now revised the manuscript further in accordance with the comments from Reviewer #1. We hope that with these amendments to our manuscript it will be suitable for publication. Please find our point by point response below.

Reviewer #1 

Tables 1-3 can be combined. 

If a nonparametric method is used, better report median (IQR) instead of mean (SD).

Answer: Table 1 and 3 has been combined to a new table 1. Table 2 is left unchanged since it has another structure and a combination would be hard to read.

We have changed to Median and IQR.

Figure 2. bars are overlapping between two groups.

Answer: The bars in Figure 2 have been separated and the color changed to grey and black.

Table 4: this table can be omitted or moved to supplemental. 

If a nonparametric method is used, better report median (IQR) instead of mean (SD).

Add percentages to treatment and sex. 

MCV superscript can be removed. 

Add “Abbreviations” to the footnote.

Answer: Previous Table 4 is now Table 3. We believe that this data should be kept in the manuscript for the best understanding of the data.

We have changed to Median and IQR.

We have added percentages to treatment and sex.

The MCV superscript has been corrected indicating statistical significance.

Abbreviations are included in the footnote.

Table 5: Why sample sizes are different for weight change, % change and reduction >=5% in control? E.g. 6m, 2 yr..

Answer: Previous Table 5 is now Table 4. Sample sizes have been corrected. Thank for the attention, we discovered a mistake in our registry.

Table 5: my calculation for the p value of reduction >=2% is 0.01. Are other pvalues accurate?

Weight % change and >=5% are secondary endpoint and p values may need to be adjusted. They can be presented in the table with other secondary endpoints.

Answer: We have only calculated for a reduction of >=5%, not 2%. Misunderstanding?

The p-value after Fisher exact test is 0.0063 but after correction with Bonferroni the p-value is 0.03 which is now stated. The other p-values have also been checked and are correct.

For best readability, we want to keep the secondary weight-related outcomes in Table 4 and have added this is the table heading.

Tables 6 and 7 can be combined.

Answer: Previous Table 6 and Table 7 have been combined and are now Table 5.

The p-values in the previous Table 8 which is now Table 6 have been corrected according to Bonferroni in accordance with the analyses with corrected numbers as for table 4.

---

## [Decision Letter · Decision Letter 2]

18 Apr 2024

“Dare to feel full” - A group treatment method for sustainable weight reduction in overweight and obese adults: A randomized controlled trial with 5-years follow-up

PONE-D-23-38830R2

Dear Dr. Holmberg,

We’re pleased to inform you that your manuscript has been judged scientifically suitable for publication and will be formally accepted for publication once it meets all outstanding technical requirements.

Kind regards,

Aleksandra Klisic

Academic Editor

PLOS ONE

Additional Editor Comments (optional):

Reviewers' comments:

Reviewer's Responses to Questions

**Comments to the Author**

1. If the authors have adequately addressed your comments raised in a previous round of review and you feel that this manuscript is now acceptable for publication, you may indicate that here to bypass the “Comments to the Author” section, enter your conflict of interest statement in the “Confidential to Editor” section, and submit your "Accept" recommendation.

Reviewer #1: All comments have been addressed

2. Is the manuscript technically sound, and do the data support the conclusions?

Reviewer #1: (No Response)

3. Has the statistical analysis been performed appropriately and rigorously? 

Reviewer #1: (No Response)

4. Have the authors made all data underlying the findings in their manuscript fully available?

Reviewer #1: (No Response)

5. Is the manuscript presented in an intelligible fashion and written in standard English?

Reviewer #1: (No Response)

6. Review Comments to the Author

Reviewer #1: All my comments are addressed.

The statistics are acceptable now.

7. PLOS authors have the option to publish the peer review history of their article (what does this mean?). If published, this will include your full peer review and any attached files.

Reviewer #1: No

---

## [Editor Report · Acceptance letter]

26 Apr 2024

PONE-D-23-38830R2 

PLOS ONE

Dear Dr. Holmberg, 

I'm pleased to inform you that your manuscript has been deemed suitable for publication in PLOS ONE. Congratulations! Your manuscript is now being handed over to our production team.

Kind regards, 

on behalf of

Dr. Aleksandra Klisic 

Academic Editor

PLOS ONE